# A Pilot Study Examining Vitamin C Levels in Periodontal Patients

**DOI:** 10.3390/nu12082255

**Published:** 2020-07-28

**Authors:** Molly-Rose Munday, Rohan Rodricks, Michael Fitzpatrick, Victoria M. Flood, Jenny E. Gunton

**Affiliations:** 1Westmead Hospital, Sydney Medical School, Faculty of Medicine and Health, The University of Sydney, Sydney, NSW 2145, Australia; mmun0421@uni.sydney.edu.au; 2Centre for Diabetes, Obesity and Endocrinology Research (CDOER), The Westmead Institute for Medical Research, The University of Sydney, Sydney, NSW 2145, Australia; 3Department of Oral Restorative Sciences (Periodontics), Westmead Centre for Oral Health, Westmead, NSW 2145, Australia; rohan.rodricks@health.nsw.gov.au; 4NSW Health Pathology, Royal Prince Alfred Hospital, Sydney, NSW 2050, Australia; michael.fitzpatrick1@health.nsw.gov.au; 5Sydney School of Health Sciences, Faculty of Medicine and Health, The University of Sydney, Sydney, NSW 2050, Australia; vicki.flood@sydney.edu.au; 6Westmead Hospital, Western Sydney Local Health District, Westmead, NSW 2145, Australia; 7Garvan Institute of Medical Research, Darlinghurst, NSW 2010, Australia

**Keywords:** vitamin C deficiencies, ascorbic acid, periodontal disease, gingivitis, adult periodontitis

## Abstract

Background: Periodontal disease is the leading cause of tooth loss worldwide. Current periodontal treatment is limited by its dependency on patients learning and maintaining good dental habits, and repeated visits to oral health physicians. Vitamin C’s role in collagen synthesis and immune function makes it important in wound healing and possibly periodontal healing. Therefore, if some patients are deficient, this may worsen patient outcomes. Methods: Patients were invited to participate following assessment and treatment at the Westmead Centre of Oral Health Periodontic Clinic, regardless of current disease stage or treatment. Adults were eligible if they gave informed consent and had current periodontal disease. Study involvement consisted of periodontal assessment and care followed by an interview and measurement of serum vitamin C and C-reactive protein (CRP). Results: A total of 6 out of 20 patients had vitamin C levels less than the institutional normal range, of whom 2 had levels <11.4 μmol/L and one <28 μmol/L. Low vitamin C was associated with higher periodontal disease stage (*p* = 0.03). Elevated CRP was found in 2/3 of people with low vitamin C and CRP was negatively correlated with vitamin C (*p* < 0.01). Vitamin C did not correlate with patient-reported fruit or vegetable consumption, but high processed meat intake was associated with lower vitamin C. Conclusion: Although a small study, this rate of vitamin C deficiency in the periodontal clinic is clinically important and correlations with disease severity and CRP suggests biological importance. This warrants further studies to assess vitamin C and whether supplementation improves periodontal outcomes, particularly in deficient subjects.

## 1. Introduction

Vitamin C (ascorbic acid) is an essential nutrient involved in a range of bodily processes such as immune function, metabolism and production of structurally sound collagen. Unlike other animals, humans, primates, guinea pigs and bats rely on nutritional sources such as citrus fruit, kiwi, tomatoes, broccoli and *Capsicum* (bell peppers) for vitamin C because they lack the enzyme required for its synthesis, L-gulono-gamma-lactone oxidase (GLO) [1]. Historically, low availability of fresh fruits and vegetables led to widespread deficiency. Severe vitamin C deficiency manifests as scurvy [2]. Symptoms of scurvy include bleeding gums, tooth loss, nausea, fatigue, eventual wound re-opening, infections, fractures, haemorrhages, delirium and death [3]. Thought to be long eliminated, scurvy has begun to re-emerge in modern populations [2,4,5]. 

Clinically, vitamin C status is usually assessed by measuring serum or plasma vitamin C levels. Blood levels correlate best with recent vitamin C intake, usually over days, except in the setting of recent supplementation. Commonly, serum vitamin C analysis is used in conjunction with questions about dietary intake of fruits and vegetables.

Periodontal disease is an umbrella term for a range of conditions affecting the periodontium. It is the result of chronic bacterial plaque build-up and inflammation resulting in damage to the underlying gingiva and alveolar bone [6,7,8]. Periodontitis is “a chronic multi-factorial chronic inflammatory disease associated with dysbiotic plaque biofilms and characterised by progressive destruction of the tooth-supporting apparatus” [9]. This leads to destruction of the alveolar bone, recession of the gums, and, if untreated, eventual tooth loss which is seen in Stage 4 disease [10]. 

The 2017 American Academy of Periodontology classification system formally incorporates clinical attachment loss around teeth, radiographical bone loss, tooth loss, probing depth, bleeding on probing, disease progression and presence of systemic diseases. At initial assessment the patient is assigned a stage and grade based on the above consensus guidelines [11]. In brief, Stage 1 is the border between simple gingivitis and periodontitis and is associated with early stages of attachment loss. Stage 2 is moderate periodontitis with damage to tooth support. Stage 3 is present when there is significant damage to the attachment apparatus of teeth and tooth loss can occur without treatment. Stage 4 is considerable damage to tooth support, often accompanied by tooth loss and difficulty with chewing of food. Grade relates to “evidence or risk of rapid progression, anticipated treatment response, and effects on systemic health” [11] and is rated A, B or C, with C the most severe.

Usual treatment involves removal of plaque and calculus deposits, sometimes in conjunction with systemic or local antibiotics and encouraging the patient to improve their daily oral hygiene practices [11]. Current periodontal treatments are effective but can be limited by their dependency on patients taking up and maintaining good dental habits.

Periodontal disease is a common chronic dental disorder that presents a large burden on both local and global societies. Together, gingivitis and periodontitis are estimated to occur in 43% of the Australian population, [12] and 47% of the US population [13], and severe periodontal disease is thought to affect more than 10% of the global population. Likely due to its inflammatory involvement, it is more common alongside a range of comorbidities including cardiovascular diseases, rheumatoid arthritis, and type 2 diabetes [14]. 

Past studies have examined vitamin C in dental patients with mild to severe gingivitis and this has been recently reviewed [3]. Overall, vitamin C levels were lower in people with gingivitis. Additionally, patients report decreased bleeding and gingival inflammation when supplemented with vitamin C [15]. An inverse relationship has also been established between the severity of necrotizing, ulcerative gingivitis and vitamin C plasma levels [16]. This is thought to be due to impaired collagen synthesis [17,18]. Therefore, vitamin C deficiency results in decreased healing capacity [19,20,21]. A number of studies such as Woelber et al. have also reported the benefits of a high quality diet for reducing gingival and periodontal inflammation [22]. Reduced bleeding was also noted when dental patients were given two grapefruits a day [23]. 

Vitamin C deficiency is more common in smokers, the elderly and people of lower socio-economic status, potentially putting those groups at increased risk of periodontal diseases [24,25]. This study aimed to examine the prevalence of vitamin C deficiency within an Australian periodontal population. We hypothesised that there would be a clinically important prevalence of vitamin C deficiency, and we tested whether simple dietary questions could predict deficiency. The background rate of deficiency in the Australian population is unknown, as is the deficiency rate in the periodontal disease population. As such, this study examined vitamin C levels in a periodontal clinic population.

## 2. Methodology

### 2.1. Ethics

This study was approved by the Western Sydney Local Health District Human Research Ethics Committee. All subjects gave informed, written consent. HREC reference number 5127–2019/ETH02304.

### 2.2. Patients

Patients were eligible to participate in this study if they were an adult attending the Westmead Centre of Oral Health (WCOH) Periodontic Clinic. Additional eligibility criteria included being able to give informed, written consent and the ability to speak and understand English (7 patients were excluded for this). Further details can be found in the patient recruitment diagram (Figure 1). Patients were approached following their consultation and/or treatment at the periodontal clinic. Three patients declined to consent upon learning that a blood test would be required. The rest were given study details and 9 declined consent at this stage. 

Consenting patients were asked about their dietary intake using a survey created by VF and JG. They then underwent a venepuncture to measure serum vitamin C and C-reactive protein (CRP) levels. Blood samples were collected, immediately wrapped in foil and placed on ice. These instructions were written on the collection request forms. As above, dental examination and assigning of stage and grade was conducted prior to study recruitment. Using a Williams Periodontal probe, six-point probing depth measurements as well as gingival recession measurements (both in millimeters) around each existing tooth were carried out for every patient. Missing teeth, tooth mobility and presence of furcations were marked. Every patient received a panoramic radiograph to assess for current alveolar bone levels around the existing dentition. Radiographic bone loss around teeth was assessed as being in the coronal third, extending to the mid third of the root or beyond (up to the apex of the tooth). The amount and distribution of supra- and subgingival plaque and calculus deposits was noted as well as the presence of bleeding and/or suppuration on gingival probing. Additional data were collected from the patients’ medical records including their age, address, smoking status and other medical disorders which were confirmed during the interview. The STROBE checklist for cross-sectional studies was used. 

Hypertension and hyperlipidaemia were assessed as present in patients who were receiving antihypertensives or lipid-lowering agents, respectively. Residential addresses were used with the ‘Socio-Economic Indexes for Australia 2016’ to estimate SES (socioeconomic score) [26]. 

### 2.3. Serum Vitamin C and C Reactive Protein (CRP) Analysis

Serum vitamin C was analysed using Hydrophilic Interaction Chromatography (HILIC) conducted at New South Wales Health Pathology, Royal Prince Alfred Hospital (RPAH). HILIC is more suitable for polar molecules such as vitamin C with good retention and separation rates. In order to increase sample stability, 100 μL of each patient sample was diluted with 100 µL of internal standard containing antioxidant tris(2-carboxylethyl)phosphine hydrochloride (TCEP, Sigma-Aldrich, Sydney, Australia). Samples were filtered through a 10 kDa centrifugal protein removal column then centrifuged for 30 min. Next, 100 μL acetonitrile (*Sigma-Aldrich*) was added to 100 µL of the supernatant, then the sample was vortexed and transferred into an HPLC vial for analysis. 

Analysis used the Dionex Ultimate 3000 UHPLC with DAD (Thermo Fisher Scientific Cat *#* 6765015, Waltham, MA, USA). The assay’s detection limit was 5μmol/L, with the normal range being 40–100 μmol/L. This was set by testing a normal group and aligns with the normal range at Monash Health in Australia. Ascorbic acid measurements were linear over an analytical range of 5–250 μmol/L. The within batch imprecision was <3% and the between batch imprecision <7%. Recovery was over 90% for the analyte and its internal standards.

CRP was measured using rate nephelometry by NSW health pathology at Westmead Hospital Institute of Clinical Pathology and Medical Research (ICPMR). CRP levels when normal are reported as <4 μmol/L, so those patients were assigned a result of 3 μmol/L. 

### 2.4. Statistical analysis 

Statistical analysis was undertaken using GraphPad Prism version 8 (San Diego, CA, USA) or SPSS Version 21 (IBM, Chicago, IL, USA). Where the data was not normally distributed (Shapiro–Wilk *p*-value < 0.05), non-parametric testing such as Independent Sample Mann–Whitney U hypothesis testing or Spearman’s correlation tests were undertaken. A *p*-value of <0.05 was considered statistically significant. Data is shown as mean ± standard deviation or where not normally distributed median (95% confidence interval).

## 3. Results

Patient demographics are reported in Table 1. Twenty patients were recruited before the dental clinic was shut down due to Covid-19. A number of comorbid conditions were reported including cardiovascular diseases, diabetes mellitus, Crohn’s disease and thyroid issues. Patients were approached regardless of their progression through treatment, or their severity of periodontal disease. Most patients were approached after their initial examination (n = 12), or during the early stages of treatment (n = 5). 

The patients were recruited after review by the treating dentist and before collection of diet information or blood tests so scores were not influenced by diet history or blood results. The 2017 world workshop consensus guidelines (published in 2018) were used to grade disease severity [11]. The patients in this study had clinically important periodontal disease; 9 of 20 patients were Stage 3, Grade B or C indicating advanced periodontal deterioration. Nine more patients were Stage 4, Grade B or C (Table 2). 

Stages and grades are as per the 2017 consensus guidelines, published in 2018. Data shows numbers of subjects in each stage/grade. Number of people in each category with vitamin C < 40 μmol/L is shown in brackets. Of the 20 patients, six had vitamin C results below the normal range (<40 µmol/L, Figure 2A, shaded area). One patient had a vitamin C of 40 μmol/L and was classified as normal. Of the 9 patients with Stage 4 periodontal disease, 5 had vitamin C <40 μmol/L. Conversely, 5 of the 6 people with low vitamin C had Stage 4 disease (Table 2). The patient with Grade 2 disease had vitamin C of 59 μmol/L, the 10 patients with Grade 3 disease had median vitamin C of 65.5 μmol/L and the 9 patients with Grade 4 periodontitis had median vitamin C of 38 μmol/L. 

People consuming the estimated average requirement of dietary vitamin C or more have levels of 38 μmol/L or higher [27]. However, most other centers use cut-offs of 28 μmol/L for insufficiency and 11.4 μmol/L for deficiency. By these criteria, 2 patients are deficient and 1 has insufficiency. Three patients below our normal range would be classified as having adequate status with levels of 29, 29 and 38 respectively. Alternately, a recent review paper suggests an adequate vitamin C for periodontal patients is 56.8 μmol/L [3]. If this is examined, 9 people had vitamin C under that level. 

Five patients had elevated CRP levels of >4 μmol/L. One of these was a patient with known Crohn’s disease whose level was 32 µmol/L, so their CRP result was excluded from further analysis. Four people with low vitamin C had elevated CRP levels (Figure 3a,b). Vitamin C and CRP were significantly inversely correlated producing a r^2^-value of 0.81 (Spearman’s correlation test) and a two tailed *p*-value of 0.0016 (Figure 3b). 

Patients each had an interview about their dietary habits around fruit and vegetable intake to see whether this could predict serum vitamin C. People with vitamin C deficiency reported a non-significant higher weekly intake of servings of fruits (median 14 (2–21) versus 7 (3–12) in people without deficiency) and no decrease in vegetable intake (median 12 (4–21) versus 12 (4–14) in people without deficiency). Interestingly, people with vitamin C < 40 μmol/L reported higher weekly servings of processed meat (4 versus 0.6, *p* = 0.024).

Using the home address for each subject, socioeconomic status (SES) was estimated as described in methods. Median SES did not differ between deficient and non-deficient groups with wide confidence intervals (median in replete group 987 (929-1094) and in deficient group 1000 (868-1045)). SES also did not differ between Stage 4 and Stage 2/3 patients (SES medians 990 and 991, respectively). 

## 4. Discussion 

This study reports that a significant proportion of patients attending the Periodontics Clinic at the Westmead Centre of Oral Health have vitamin C deficiency. Although a small sample size, 6 of 20 people having results below the normal range is a startling statistic that warrants further research. Patients with low vitamin C were notified of their test results, and 500 mg per day of vitamin C was recommended. When considered in the context of scurvy, periodontal disease shares a number of similarities such as excessive gum bleeding and tooth loss, therefore it is likely that some periodontal patients meet the requirements for scurvy diagnosis. Because low vitamin C correlated with more advanced periodontal disease and with increased CRP, a measure of systemic inflammation, it is likely that this finding is biologically important.

CRP is recognised to be elevated in some patients with periodontal disease, and in those people, treatment is associated with lowering of CRP [28]. 

The next step may be to conduct a randomised placebo-controlled trial, with the hypothesis that vitamin C will assist periodontal healing in people who are deficient. If whole-body vitamin C status is adequate, then it seems unlikely that supplementation would benefit that particular individual. Additionally, as our dental service sees tertiary referral cases, the deficiency proportion in our patients is likely to be more severe than seen in normal dental practices. Therefore, studies in community-based dental clinics in Australia and other countries are also of interest. However, it is worth noting that the 3 in 10 rate of results below the normal range rate in this study is lower than seen in our High-Risk Foot Ulcer Clinic patients (50%, unpublished data).

Although research has been conducted concerning the relationship between vitamin C and periodontal disease, there is a lack of studies examining the prevalence of vitamin C deficiencies within the Australian periodontal population and general Australian population. This study provides pilot data for the prevalence of such deficiencies. There are a number of limitations to this study, the most obvious being sample size. Study recruitment had to be halted when the dental clinic was closed due to COVID-19. Secondly, recruitment was limited to the WCOH Periodontic Clinic and non-tertiary centres may find lower deficiency rates. Thirdly, as this was a cross-sectional study, it was not able to assess whether deficiency affected periodontal healing, or whether treatment improved healing. Additionally, the short survey of diet may be an imprecise indicator of vitamin C intake, and a different screen of diet (such as 24 h recalls) may provide further insights into the association of dietary intake and vitamin C status [29].

## 5. Conclusions 

In conclusion, vitamin C deficiency was prevalent at an unacceptable rate in our periodontal clinic. Deficiency correlated with more severe periodontal disease, and with increased systemic inflammation. We recommend consideration of testing of vitamin C for patients with significant periodontal disease, particularly if other features of scurvy are present such as bruising or corkscrew hairs, or if periodontal healing does not progress as expected. Further research should be conducted into the relationships between vitamin C and periodontal disease, and a randomised, controlled trial would be ideal.

## Figures and Tables

**Figure 1 nutrients-12-02255-f001:**
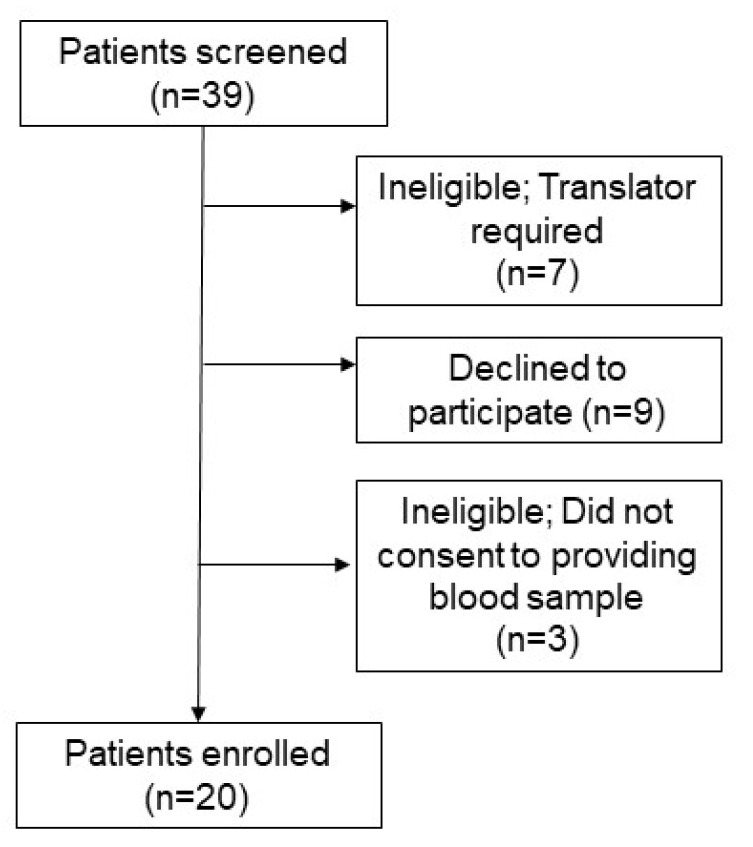
Patient recruitment flow chart. Patients attending the dental clinic were screened for eligibility to participate.

**Figure 2 nutrients-12-02255-f002:**
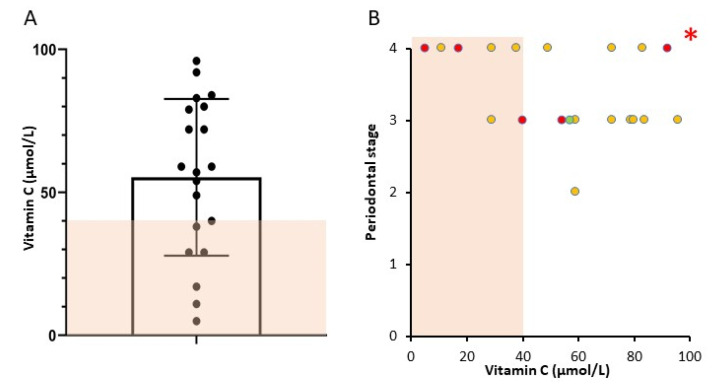
Serum vitamin C and periodontal stage. (**A**) Individual vitamin C results for each patient. The shaded area indicates results below the assay normal range. One patient returned a serum vitamin C of 40 µmol/L (normal 40–100 µmol/L) and was classified as normal. (**B**) Periodontal stage was evaluated by the treating dentist (RR) prior to blood collection for vitamin C. Symbol fill colour indicates grade of periodontal disease (A = green, B = yellow, C = red). * = *p* = 0.03.

**Figure 3 nutrients-12-02255-f003:**
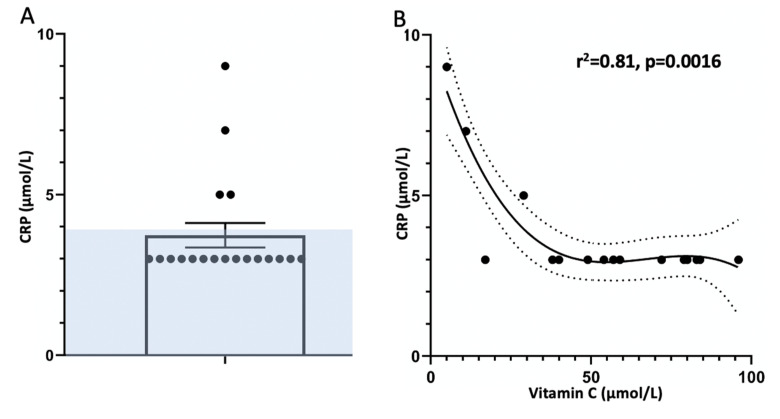
CRP (C-reactive protein) in periodontal patients. (**A**) Individual CRP results, excluding the patient with Crohn’s disease whose level was 32 μmol/L. Normal CRP was reported as <4 µmol/L so these were entered as 3 µmol/L. (**B**) Correlation between vitamin C and CRP. r^2^ = 0.81, *p =* 0.0016.

**Table 1 nutrients-12-02255-t001:** Patient demographics.

Patient Data	Numbers
Number	20
Male/Female	10/10
Age (years)	65 ± 9
Average SES Score *	990 (939–1022)
Smokers	4 (1:5)
Hypertension	5 (1:4)
Hyperlipidaemia	3 (3:20)
Diabetes	4 (1:5)
Crohn’s disease	1 (1:20)
Thyroid disease	3 (3:20)

Information was obtained from patient records and/or dietary survey. * Socioeconomic score (SES) was calculated based using the patient’s address [26]. 1000 is the Australian median. Data shows mean±standard deviation or median (95% CI).

**Table 2 nutrients-12-02255-t002:** Patient periodontal characteristics.

Periodontal Staging	Grade A	Grade B	Grade C
**Stage 1**	0	0	0
**Stage 2**	0	1 (0 deficient)	0
**Stage 3**	1 (0 deficient)	7 (1 deficient)	2 (0 deficient)
**Stage 4**	0	7 (3 deficient)	2 (2 deficient)

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
