# Peer review of "A Pilot Study Examining Vitamin C Levels in Periodontal Patients"

_nutrients, 2020, doi:10.3390/nu12082255_

Round 1

Reviewer 1 Report

The study by Munday et al examines an interesting question and, if the findings prove to be replicated, this may be something to explore further through an RCT, as they suggest. They manuscript is well written and the analyses conducted appear appropriate, but my significant concern is the threshold used to define vitamin C deficiency: <40 umol/L serum ascorbic acid. My understanding is that serum ascorbic acid concentrations are considered to be adequate if >28 umol/L, suboptimal if between 11 and 28 umol/L, and deficient if <11 umol/L (see, for example, https://academic.oup.com/ajcn/article/90/5/1252/4598114). As such, I cannot further comment on the suitability of this manuscript for publication until analyses using the traditional definition of deficiency are carried out.

Author Response

Thank you, we have incorporated additional text in the methods section about the assay and its normal range. It was established some years ago according to the usual method of testing a group of healthy people on no regular medications. Our normal range is very similar to that of Monash in Australia who use similar methodology, and may relate to the HILIC methodology (Lines 136-138 in clean version, 157-159 tracked version). 

We made a point of writing on every sample collection form ‘Wrap in foil and place on ice immediately” which we hope improved the vitamin C retention of our samples, decreasing the risk of ‘false positive’ low results.

We have added to the results section using the cutoff of 28μmol/L (3 subjects still deficient), and also using the paper recommended for addition to the manuscript by reviewer 2. This is a review of periodontal disease and vitamin c and uses cell-vitamin C content to recommend an optimal serum content which it suggests is 56.8. This is now lines 235-239 in the tracked version / 203-207 in the clean version. Ideally, an RCT would help to establish whether there is a threshold for vitamin C and periodontal disease.

Reviewer 2 Report

Title: “The Prevalence of Vitamin C Deficiency in Periodontal Patients” (nutrients-831801)

The authors present a study in which they perform an assessment of vitamin C levels in periodontal patients and explore this association and related factors. Although it is an interesting topic, it has been performed in a small sample, and this reviewer has found several shortcomings in the manuscript that should be addressed.

Comment to the authors

Title page and abstract:

  • Methods section of the abstract should better focus on the type of patients and describe the methodologies used instead of the place where the study took place.
  • Please ensure that the keywords are MeSH terms. Use the appropriate terms as specified in MeSH for better indexation of the manuscript.
  • It is indeed a small study, and therefore it may be noted in the title as “pitot study”.

Introduction

  • Latin terms such as Capsicum should be written in italics.
  • Last sentence of the first paragraph is a sort of study objectives, it should be placed at the end of introduction as study objectives usually are.
  • Please provide accepted and focused definitions for periodontal disease, gingivitis and periodontitis based on the recent international classification of 2018. The used definition of periodontitis involves only microbiological cause, when the host response also plays a major role.
  • This section should include is there is any previous evidence concerning vit c levels and periodontitis, as rationale for the study (as they use recent evidence as: Vitamin C and Its Role in Periodontal Diseases - The Past and the Present: A Narrative Review. Van der Velden U. Oral Health Prev Dent. 2020;18(2):115-124).

Materials and Methods

  • Funding statements should go at the end of the manuscript, please follow the templates for the journal.
  • Provide a citation for the method or the database used for the estimation of socioeconomic status.
  • Please comment on the study design and follow an appropriate guideline for the manuscript preparation and specify this in the methods section. In this case, STROBE guidelines are the appropriate for observational studies.
  • What was the case definition of periodontitis? There is also no reference on how the periodontal examination was performed or which periodontal clinical variables were gathered.
  • Did the study follow any calibrated and internationally accepted questionnaire to assess fruit and vegetable intake?
  • In should not be named CONSORT diagram since this study is not a clinical trial and therefore it does not follow the CONSORT guidelines.
  • Please provide complete references of the commercial products listed in the study i.e. “Thermo Fisher Scientific” and the statistical software used.
  • Please provide citations on the technique described for vitamin C assessment.
  • Statistical tests: How was normality of the data assessed? If data was non-normally distributed, maybe it should be shown as median and interquartilic range instead.

Results and discussion

  • Please follow the table format required by the template of the journal.
  • A number of variables are presented in table 1 that have not been described how were gathered and their definitions in methods section.
  • Could also mean values of vit C compared between each periodontitis stage group? Apart from using deficiency as a binary variable.
  • I suggest presenting the data in figures also in tables to see complete values and their significances (figures 2 and 3).
  • Should the R2 of the correlation be negative? Since CRP levels decrease when Vit C levels increase (inversely correlated).
  • Conclusions are too extensive. Authors should only conclude the results of the study regarding the initial objectives proposed. Sentences where they hypothesize, justify or make recommendations should be better described in discussion.

Author Response

Thankyou, we have renamed the paper “A Pilot Study Examining the Prevalence of Vitamin C Deficiency in Periodontal Patients”

The abstract has been rewritten to increase the information in the methods but to maintain the word count.

The keywords have been amended to be MeSH headings and are now separated by semicolons.

Introduction

Latin terms- in Australia capsicum is the common name for what is called bell pepper in the USA. We have italicized capsicum as suggested.

We have moved the last sentence of the first paragraph to the end of the introduction as suggested.

Thank you, we have added text around the consensus guideline definitions and removed some of the wording around inflammation. Rather than trying to reproduce the information in the guidelines, we have briefly discussed gingivitis and periodontitis and refer to the guidelines which provide a full summary. This was text in the results, which is now in the introduction on page 3 (track changes, but page 2 of the clean version).

Text has been added as suggested in the introduction about vitamin C and dental diseases, including the recent review paper by Van der Veldena, thank you.

Methods

The funding statement is now only at the end.

A reference for the SES data has been added, thank you. The website allows input of postcode (same as zipcode in USA) or specific address and then gives the estimated SES.

The case definition of periodontitis was as per the 2017 consensus (published 2018).

The questionnaire was developed by our Professors of Dietetics (VF) and Medicine (JG) to see if some simple questions could predict the vitamin C result. As stated in the paper, it did not predict the outcome, possibly because people were aware of the purpose of the study. I’ve appended it to the end of this response in case you’d like to see it (but please note the formatting has gone off with the cut and pasting).

Consort diagram; thank you, we have renamed the figure “Patient recruitment flow-chart”.

Information for the commercial products has been added.

We have expanded upon the methods for vitamin C collection and measurement.

Results

The tables have been formatted as per the guidelines in https://www.mdpi.com/journal/nutrients/instructions

The variables are as described in the patient’s medical records. Hypertension was marked as present for people taking antihypertensives and hyperlipidaemia in people on lipid lowering agents.

We have moved the definitions of periodontal disease from the results section up to the introduction on page 2.

Vitamin C for each periodontitis group is now shown on page 11, and says “The patient with Grade 2 disease had vitamin C of 59μmol/L, the 10 patients with grade 3 disease had median vitamin C of 65.5μmol/L and the 9 patients with grade 4 periodontitis had median vitamin C of 38μmol/L. Mean vitamin C results were 59, 65 and 44 for grades 2, 3 and 4.”

We have incorporated (on page 12) sentences in the results text using reviewer 1’s suggested cutoff of 28 and also the paper you suggested by Van der Veldena which recommends 56.8μmol/L.

I suggest presenting the data in figures also in tables to see complete values and their significances (figures 2 and 3).” All individual patient values are already shown in figures 2 and 3. We’ve added the text as discussed above to show median vitamin C values per stage of dental disease.

“Should the R2 of the correlation be negative? Since CRP levels decrease when Vit C levels increase (inversely correlated).” No, because it’s r-squared, and squaring any negative number results in a positive. The un-squared r is = -0.898.

Results

“Conclusions are too extensive. Authors should only conclude the results of the study regarding the initial objectives proposed. Sentences where they hypothesize, justify or make recommendations should be better described in discussion.”  We have limited the comments about the diet survey to the discussion as suggested.

Round 2

Reviewer 1 Report

While the paper is improved, the authors state that "Some other centres use a cut-off of 28μmol/L of vitamin C to indicate insufficiency." This is a serious understatement. Vitamin C deficiency is defined as less than 11.4 umol/L internationally (see this Australian paper for an example of that threshold being used there: https://pubmed.ncbi.nlm.nih.gov/30177219/. In addition, the authors can read the Institute of Medicine report on how the DRIs for vitamin C were derived here: https://www.ncbi.nlm.nih.gov/books/NBK225478/#ddd0000027).

I can certainly entertain that the threshold for periodontal disease may be higher, but the authors should not use the definition of less than 40 umol/L as deficient (or even insufficient) because this is not what the nutrition community defines as vitamin C deficiency. Perhaps the authors can split their study population into quartiles or tertiles of blood ascorbate and do the analyses accordingly without making references to vitamin C deficiency - or even analyze the association in a continuous fashion with linear regression. That may provide a clearer picture of where along the serum vitamin C continuum periodontal disease is more likely to occur. But making statements such as 30% of the population being vitamin C deficient is incorrect, since this is not the case. 

Reviewer 2 Report

Manuscript Review Comments

Title: “A pilot study examining the Prevalence of Vitamin C Deficiency in Periodontal Patients” (nutrients-831801.R1)

Comment to the authors

The authors have correctly addressed most of the comments and the quality of the article has improved. However, some minor issues still remain:

  • I understand that it may be the common term used in Australia, but since this is a scientific paper the Latin term should be the correct one to use. Since it is the taxonomy genus, it should be written in italics and with first letter capital (Capsicum).
  • Please pay attention to the acronym use, since some of them are described at first mention but never used again in the manuscript, i.e. RBL.
  • Authors have not answered to my previous query: Please comment on the study design and follow an appropriate guideline for the manuscript preparation and specify this in the methods section. In this case, STROBE guidelines are the appropriate for observational studies.
  • If the case definition of periodontitis was the one from 2017 consensus, it should be stated in methods. Also, a more detailed description on how the periodontal examination was performed is necessary (sites per tooth were probed, was does the variables “bone” refer to, if any periodontal indexes were calculated cite them etc.).
  • Commercial references are still incomplete (SPSS should be IBM, Chicago, IL, USA) and some of them are not detailed. Please be thorough and do not only refer to the examples I gave in my previous comment (i.e. TCEP antioxidant and the rest of the reagents).
  • Authors have not answered to my previous query: Did the study follow any calibrated and internationally accepted questionnaire to assess fruit and vegetable intake?
  • Statistical tests: Usually normality test such as Shapiro-Wilk or Kolmogorov-Smirnov are more commonly used instead of the measures proposed by the authors.
